# SMAD4 and TGFβ are architects of inverse genetic programs during fate determination of antiviral CTLs

Karthik Chandiran[1], Jenny E Suarez-Ramirez[1], Yinghong Hu[2], Evan R Jellison[1], Zeynep Ugur[1], Jun Siong Low[3], Bryan McDonald[4], Susan M Kaech[4], Linda S Cauley[1]*

[1]Department of Immunology, University of Connecticut Health Center, Farmington, United States; [2]Department of Microbiology and Immunology, Emory Vaccine Center, Emory University, Atlanta, United States; [3]Department of Immunobiology, Yale University School of Medicine, New Haven, United States; [4]NOMIS Center for Immunobiology and Microbial Pathogenesis, Salk Institute for Biological Studies, La Jolla, United States

**Abstract** Transforming growth factor β (TGFβ) is an important differentiation factor for cytotoxic T lymphocytes (CTLs) and alters the expression levels of several of homing receptors during infection. SMAD4 is part of the canonical signaling network used by members of the transforming growth factor family. For this study, genetically modified mice were used to determine how SMAD4 and TGFβ receptor II (TGFβRII) participate in transcriptional programming of pathogen-specific CTLs. We show that these molecules are essential components of opposing signaling mechanisms, and cooperatively regulate a collection of genes that determine whether specialized populations of pathogen-specific CTLs circulate around the body, or settle in peripheral tissues. TGFβ uses a canonical SMAD-dependent signaling pathway to downregulate Eomesodermin (EOMES), KLRG1, and CD62L, while CD103 is induced. Conversely, in vivo and in vitro data show that EOMES, KLRG1, CX$_3$CR1, and CD62L are positively regulated via SMAD4, while CD103 and Hobit are downregulated. Intravascular staining also shows that signaling via SMAD4 promotes formation of long-lived terminally differentiated CTLs that localize in the vasculature. Our data show that inflammatory molecules play a key role in lineage determination of pathogen-specific CTLs, and use SMAD-dependent signaling to alter the expression levels of multiple homing receptors and transcription factors with known functions during memory formation.

*For correspondence:
lcauley@uchc.edu

Competing interest: The authors declare that no competing interests exist.

## Editor's evaluation

This manuscript provides important information concerning the impact of TGFB and SMAD factors on antiviral CD8 T cell differentiation and memory formation. The new and interesting data from in vivo performed experiments provide solid evidence in support of their claims. Overall, this is an important study with an impressive amount of data. The only weak point is the clarity of the overall goal/message of the study as stated in the introduction.

## Introduction

Cytotoxic T lymphocytes (CTLs) make important contributions to protective immunity by eliminating cells that contain intracellular pathogens, however rapid mobilization to infected tissues is a decisive factor in immune control. Defining how activated CTLs differentiate into functionally specialized

subsets of memory CD8 T cells is an important priority for vaccine development. Our study sheds new light on this important process by revealing how two intersecting signaling pathways control a major bifurcation in diffrentiation pathway of newly activated CTLs. Multiple subsets of CTLs can be distinguished using surface markers that influence their effector functions, survival properties, and capacity to localize in specific tissues. The major subsets include central memory CD8 T cells ($T_{CM}$) that use the bloodstream to circulate through secondary lymphoid organs (SLOs), as well as some effector memory CD8 T cells ($T_{EM}$) and tissue resident memory CD8 T cells ($T_{RM}$) that distribute to peripheral tissues. Efforts to understand how this heterogeneity develops have inspired alternative models of CD8 T cell differentiation (*Montacchiesi and Pace, 2021*). Some studies suggest that antigen-specific CTLs follow a linear differentiation pathway by transitioning between subsets, while others favor a branching model of differentiation whereby multiple subsets of CTLs arise from pathways that diverge soon after antigen stimulation (*Bannard et al., 2009*). Although many studies show that environmental cues influence the transcriptional programs of pathogen-specific CTLs, the pathways that promote commitment to specific lineages are poorly defined. Our study focuses on a network of signaling pathways that are used by members of the transforming growth family (TGF). We show that two distinct signaling pathways influence the transcriptional programs of newly activated CTLs to provide customized responses during infections with different types of pathogens. The target genes are transcription factors and adhesion molecules that determine how CTLs distribute in different tissues.

Pathogen-specific CTLs express many homing receptors that are dynamically regulated during infection. Under resting conditions, SLOs are surveyed by naive CD8 T cells that search for antigen-presenting cells bearing pathogen-derived peptides. Circulating leukocytes enter resting lymph nodes from wide blood vessels known as high endothelial venules (HEVs) (*Mionnet et al., 2011*). CD62L (L-selectin) is an adhesion molecule that interacts with carbohydrates expressed on the surface of activated endothelial cells (*Steeber et al., 1998*). Naive CD8 T cells leave the bloodstream after CD62L initiates an adhesion cascade that results in transendothelial migration (*Ding et al., 2000*). Inside the lymph node, signals from antigen receptor and costimulatory molecules induce clonal expansion and CD62L is replaced with adhesion molecules that facilitate migration to the site of infection.

Cytokines that are released in infected tissues influence the transcriptional programs of activated CTLs. During the acute phase of infection, interleukin (IL)-12 works with other inflammatory molecules to promote formation of short-lived effector T cells ($T_{EFF}$) that are primed for rapid effector functions (*Chowdhury et al., 2011*; *Cui et al., 2009*; *Keppler et al., 2012*). Some pathogens elicit robust inflammatory responses and generate large populations of terminally differentiated $T_{EFF}$ cells that express killer cell lectin-like receptor G1 (KLRG1) (*Obar et al., 2011*; *Plumlee et al., 2013*), which is a membrane-bound adhesion molecule with an immunoreceptor tyrosine-based inhibitory motif in the cytoplasmic tail (*Ito et al., 2006*; *Rosshart et al., 2008*; *Tessmer et al., 2007*). Although CTLs transiently express KLRG1 during antigen stimulation (*Herndler-Brandstetter et al., 2018*; *Joshi et al., 2007*), stable expression identifies CTLs that do not convert to a conventional memory phenotype (*Dominguez et al., 2015*; *Plumlee et al., 2013*). Most $T_{EFF}$ cells disappear as the infection resolves, leaving residual populations of memory CD8 T cells in the circulation and peripheral tissues. Intravascular (IV) staining shows that some surviving CTLs maintain KLRG1 expression inside the vasculature during the memory phase of infection (*Hu et al., 2015*). Similar populations of KLRG1+ CTLs that were detected after infections with other pathogens are referred to as long-lived effector cells (LLECs), or terminal effector memory CD8 T cells ($T_{TEM}$) (*Milner et al., 2020*; *Renkema et al., 2020*). As inflammation resolves CD62L is reexpressed on $T_{CM}$ cells, while CD69+ $T_{RM}$ cells remain in the infected tissues (*Mueller et al., 2013*). How these subsets contribute to protective immunity varies according to the type of pathogen.

TGFβ is an immunosuppressive cytokine with multiple functions during CD8 T cell differentiation. Members of the TGF cytokine family elicit cellular responses via both canonical and noncanonical signaling pathways (*Derynck and Budi, 2019*). Canonical signals are mediated by a cascade of structurally related molecules known as small (Sma) and mothers against decapentaplegic (Mad)-related (SMAD) proteins (*Massagué, 2012*). SMAD4 is an adaptor molecule that chaperones phosphorylated receptor-activated SMAD proteins (R-SMADs) into the nucleus to modulate gene expression. We previously used genetically modified mice to explore how pathogen-specific CTLs distribute in the lungs during infection with influenza A virus (IAV) and discovered that several adhesion molecules were regulated via the canonical SMAD signaling cascade (*Hu et al., 2015*). Extensive phenotypic changes

indicated that loss of signaling via SMAD4 caused defects during formation of $T_{EFF}$ and $T_{CM}$ cells (*Cao et al., 2015*; *Hu et al., 2015*). To further define that mechanism that was responsible for these changes, we used multiple strains of genetically modified mice to explore how the SMAD signaling network influences genetic programming of pathogen-specific CTLs during infection. Our data show that SMAD4 and TGFβ receptor II (TGFβRII) use alternative signaling mechanisms to alter the expression levels of the same genes in opposite directions. The target genes include the T-box transcription factor EOMES, which is normally expressed during formation of $T_{CM}$ cells and downregulated by TGFβ during $T_{RM}$ development (*Intlekofer et al., 2005*; *Mackay et al., 2015*). We show that during transcriptional programming of newly activated CTLs, SMAD4 positively regulates EOMES expression and induces phenotypic changes that indicate progression toward a central memory phenotype, while formation of $T_{RM}$ cells is inhibited.

## Results

### SMAD4 modifies homing-receptor expression in the absence of TGFβ

Several groups have used Cre-lox recombination to prevent expression of either SMAD4 (S4KO) or TGFβ receptor II (TR2KO) in peripheral CD8 T cells and found altered homing-receptor expression on antigen-experienced CTLs (*Cao et al., 2015*; *Hu et al., 2015*; *Igalouzene et al., 2022*; *Wu et al., 2021*; *Zhang and Bevan, 2013*). For these studies, different promoters were used to induce gene ablation which may account for some variations between the results. CD4-Cre, CD8-Cre, and the proximal Lck promoter (dLck-Cre) are all expressed during early stages of thymic development. When these promoters were to prevent SMAD4 expression in peripheral CD8 T cells, investigators reported modified effector responses after T cell activation (*Cao et al., 2015*; *Liu et al., 2022*; *Wu et al., 2002*). Since our goal was to avoid altering naive CD8 T cells during thymic development, we used the distal Lck promoter (dLck) to prevent SMAD4 expression. When we analyzed pathogen-specific CTLs during IAV infection, very few SMAD4-deficient CTLs expressed KLRG1 or CD62L, while abnormal CD103 expression was detected on CTLs in the spleen (*Hu et al., 2015*). These changes were unexpected, as CTLs displayed a reciprocal phenotype after the TGFβ receptor was ablated (*Hu et al., 2015*). We found that the mutation primarily influenced homing-receptor expression, since SMAD4-deficient CTLs proliferated and expressed IFNγ and TNFα at similar levels as wildtype CTLs (*Hu et al., 2015*).

SMAD4 and TGFβRII are components of an interconnected signaling network. To further understand how these molecules control homing-receptor expression, we intercrossed S4KO and TR2KO mice to produce CTLs with both mutations (S4TR2-DKO). The mice were further crossed with OTI mice that express a transgenic antigen receptor that is specific for a peptide (SIINFEKL) derived from chicken ovalbumin (OVA) presented on H-2K$^b$ (*Hogquist et al., 1994*). To analyze homing-receptor expression during infection, the mice were infected with recombinant pathogens (X31-OVA and LM-OVA) that express the SIINFEKL peptide derived from chicken OVA (*Jenkins et al., 2006*; *Pope et al., 2001*).

For this study, two pathogens were used to compare homing-receptor expression under different inflammatory conditions. To elicit a mild inflammatory response in the lungs, mice were infected with IAV (X31-OVA) as neuraminidase activates TGFβ (*Schultz-Cherry and Hinshaw, 1996*). Other mice were infected with *Listeria monocytogenes* (LM) to elicit an inflammatory response that supports formation of CTLs that express KLRG1 (*Plumlee et al., 2015*). To monitor changes in homing-receptor expression during infection, congenically marked OTI-Ctrl (lack Cre), OTI-TR2KO; OTI-S4KO and OTI-S4TR2DKO cells were transferred to B6 mice 48 hr before infection. On different days post infection (dpi), donor cells in the lungs and spleens were analyzed for CD103, KLRG1, and CD62L expression (*Figure 1A, B*). Statistical comparisons are shown in the supplementary data (*Supplementary file 2*). Prior studies show that KLRG1 is negatively regulated by TGFβ (*Sanjabi et al., 2009*; *Schwartz-kopff et al., 2015*). Since latent-TGFβ can be activated by viral neuraminidase (*Schultz-Cherry and Hinshaw, 1996*), only small numbers of OTI-TR2Ctrl cells expressed KLRG1 after IAV infection, whereas most OTI-TR2KO cells expressed KLRG1 at high levels (*Figure 1A*). In contrast, there was little difference between percentages OTI-TR2Ctrl and OTI-TR2KO that expressed KLRG1 during the early stages of infection with LM-OVA (*Figure 1B*), indicating minimal influence of TGFβ during bacterial infection. As the infection progressed, some OTI-TR2KO cells reexpressed CD62L indicating conversion to $T_{CM}$ phenotype. The OTI-S4KO cells mostly lacked KLRG1 and CD62L expression after

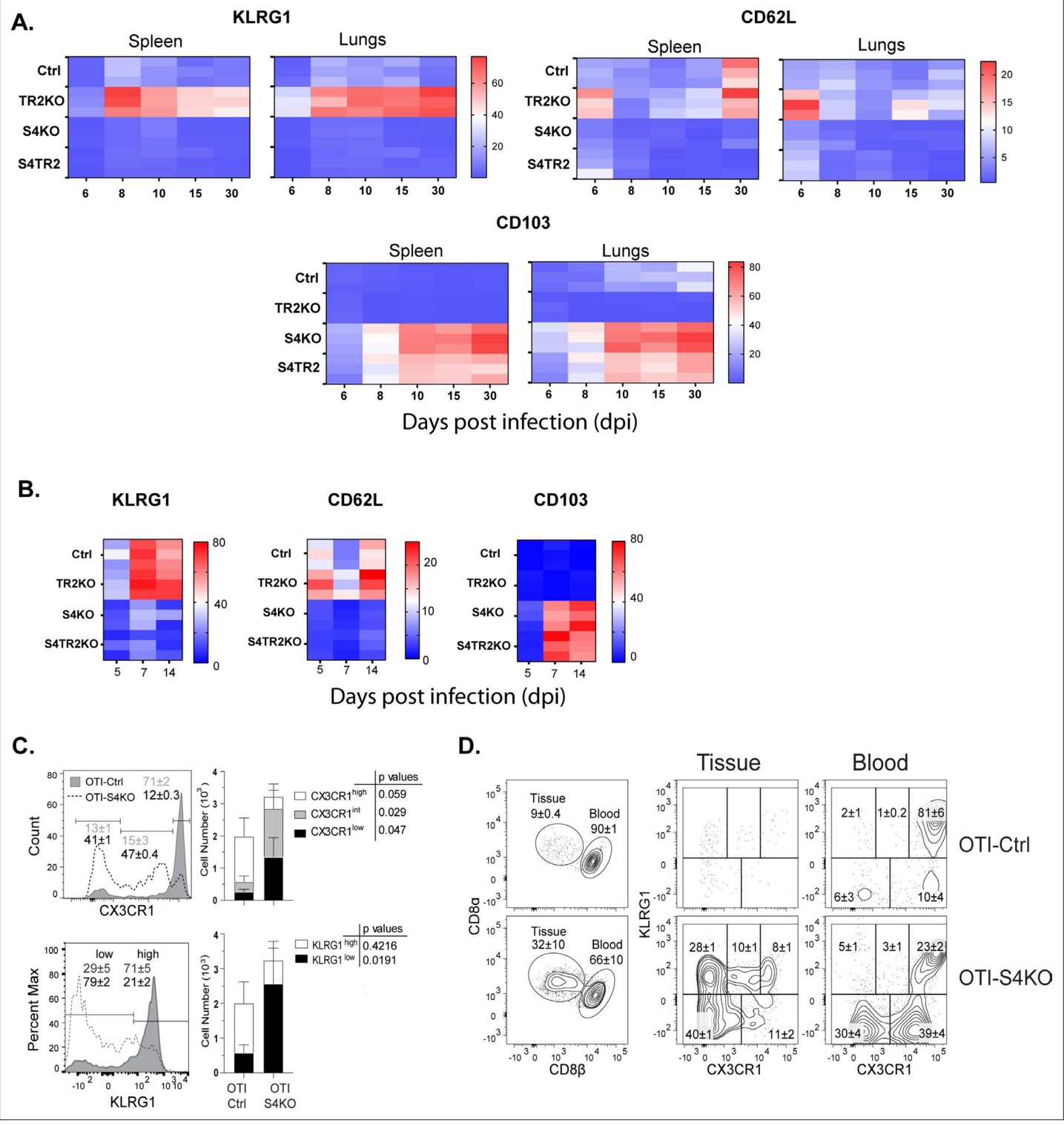

**Figure 1.** SMAD4 ablation alters homing-receptor expression without transforming growth factor β (TGFβ) receptor II. (**A, B**) Naive donor cells from OTI-Ctrl, OTI-TR2KO, OTI-S4KO, and OTI-S4TR2-DKO were transferred to B6 mice before infection. Heatmaps show percentages of donor cells that expressed KLRG1, CD103, CD62L, and CD127 on different day post infection. Data are means ± standard deviation (SD) (*n* = 3/group). Two independent experiments gave similar results. p values were calculated using two-way ANOVA and statistical comparisons were made using Tukey's multiple comparisons test. (**A**) Donor cells in the lungs and spleens after i.n. infection with X31-OVA. (**B**) Donor cells in the spleens after i.v. infection with LM-OVA. (**C, D**) IV staining shows OTI-S4KO and OTI-S4Ctrl cells in the lungs at 34 dpi with LM-OVA given by i.n. inoculation. Data are means ± standard deviation (SD) (*n* = 3-4/group). Two independent experiments gave similar results. p values were calculated using Student's *t* tests (**C**) Histograms

*Figure 1 continued on next page*

Figure 1 continued

(percentages) and bar graphs (cell numbers) show all donor cells from OTI-S4KO (dashed) and OTI-S4Ctrl mice (continuous line). (**D**) Contour plots show donor cells in the bloodstream (right panel) and lung tissue (left panel).

The online version of this article includes the following figure supplement(s) for figure 1:

**Figure supplement 1.** Naive OTI-S4TR2Ctrl (gray shading) and OTI-S4TR2KO cells (dotted line) were labeled with carboxyfluorescein succinimidyl ester (CFSE) and transferred to C57BL/6 mice before infection with X31-OVA.

**Figure supplement 2.** Mixed donor cells (OTI-S4KO and OTI-S4Ctrl) were transferred to C57BL/6 mice before infection with LM-OVA.

**Figure supplement 3.** Weight changes in S4TR2-Ctrl (filled squares) and S4TR2-DKO mice (open squares) after primary infection with X31-OVA (H3N2 serotype).

**Figure supplement 4.** Weight changes in S4TR2-Ctrl (filled squares) and S4TR2-DKO mice (open squares) after secondary infection with WSN-OVA$_I$ (H1N1 serotype).

infection with both pathogens, indicating a defect during formation of $T_{EFF}$ and $T_{CM}$ cells (*Cao et al., 2015*; *Hu et al., 2015*). The OTI-S4KO and OTI-S4TR2-DKO cells expressed KLRG1 and CD62L at similar levels, whereas the increase in CD103 expression was slightly more pronounced on S4KO cells than OTI-S4TR2-DKO cells. p values are shown in *Supplementary file 2*. These data are consistent with other studies and show that SMAD4 regulates activated CTLs via a TGFβ-independent pathway (*Igalouzene et al., 2022*; *Wu et al., 2021*). We also compared cell proliferation during IAV infection using carboxyfluorescein succinimidyl ester (CFSE) dilution and 5-bromo-2-deoxyuridine (BrdU) incorporation (*Figure 1—figure supplement 1*). The data show that OTI-S4TR2DKO and OTI-S4TR2Ctrl cells proliferated at similar rates, indicating normal T cell priming.

We previously used IV staining to analyze CTLs during IAV infection and found some KLRG1$^+$ CTLs inside the blood vessels of the lungs >40 dpi (*Hu et al., 2015*). Since very few SMAD4-deficient CTLs expressed KLRG1 during infection with LM-OVA, we investigated whether these cells were located in the vasculature or peripheral tissues. Fractalkine ($CX_3CL_1$) is a chemoattractant with adhesive properties (*Ostuni et al., 2020*) that alters CD8 T cell migration. Prior studies indicate that CTLs in the vasculature express the fractalkine receptor ($CX_3CR1$) at high levels (*Nishimura et al., 2002*). Several groups used reporter mice to monitor changes $CX_3CR1$ expression during infection with different pathogens and found that terminally differentiated $T_{EFF}$ cells expressed KLRG1 and $CX_3CR1$ at high levels (*Gerlach et al., 2016*; *Jung et al., 2000*). To explore whether $CX_3CR1$ is regulated via SMAD4, we crossed the $CX_3CR1^{GFP+}$ reporter with OTI-S4KO mice for transfer experiments. Naive OTI-Ctrl and OTI-S4KO cells were mixed (1:1 ratio) and transferred to B6 mice before infection with LM-OVA. After 30 days, donor cells were analyzed for KLRG1 and GFP (surrogate for $CX_3CR1$) expression (*Figure 1C, D* and *Figure 1—figure supplement 2*). Fluorescently conjugated antibodies (CD8β) were injected 3 min before sacrifice, to distinguish CTLs in the blood vessels and peripheral tissues. The majority of OTI-Ctrl cells in the lungs expressed KLRG1 and GFP at high levels (overlaid histogram and bar graph) (*Figure 1C*) and were primarily located inside the blood vessels (*Figure 1D*). Conversely, OTI-SKO cells expressed KLRG1 and GFP at intermediate/low levels (*Figure 1C*), and some cells were located in the tissue (*Figure 1D*). Similar distributions of donor cells were found in the red and white pulp of the spleen (*Figure 1—figure supplement 2*). Since small numbers of SMAD4-deficient CTLs expressed KLRG1 and CX3CR1 at high levels, the data indicate that SMAD4 acts as a catalyst for terminal differentiation.

Rodents lose large percentages of their body weight during IAV infection and the rate of recovery to normal size can be used to assess the severity of disease. After IAV infection, S4KO mice experienced more severe disease than the controls (*Hu et al., 2015*). Since OTI-S4KO and OTI-S4TR2-DKO cells displayed similar phenotypes during infection, we used weight loss to measure of protective immunity. S4TR2-DKO and S4TR2-Ctrl mice were infected with X31-OVA and weight changes were recorded daily (*Figure 1—figure supplement 3*). As expected, the S4TR2-DKO mice lost more weight and recovered from infection more slowly than the controls. After 30 days, the immune mice (and uninfected controls) were challenged with different strain of IAV (WSN-OVA$_I$) (*Figure 1—figure supplement 4*). None of the immune mice experienced substantial weight loss after infection with WSN-OVA$_I$, indicating that protection against reinfection was not dependent on SMAD4 or TGFβ.

## SMAD4 modifies homing-receptor expression independently of R-SMAD2/3

The TGF family includes two groups of cytokines, that signal via alternative branches of the SMAD cascade. TGFβ and activins signal via R-SMAD2/3, while bone morphogenic proteins and related growth differentiation factors signal via R-SMAD1/5/8 (*Fink et al., 2003*; *Takimoto et al., 2010*). SMAD4 facilitates signaling by both groups of cytokines and chaperones complexes of phosphory-lated R-SMADs into the nucleus for gene regulation (*Massagué et al., 2005*). Since we found that SMAD4 was required for formation of $T_{CM}$ cells but not $T_{RM}$ cells, we used quantitive polymerase chain reaction (qPCR) to compare SMAD4 expression in these subsets and found only a moderate differ-ence in transcript levels (*Figure 2—figure supplement 1*).

When CTLs are stimulated with TGFβ, the receptor phosphorylates R-SMAD3 and induces CD103 expression (*Yang et al., 1999*). Prior studies show some redundancy within the SMAD signaling network. After cancer cells were stimulated with TGFβ, R-SMAD2 and R-SMAD3 were phosphorylated and formed heterodimers that entered the nucleus in the absence of SMAD4 (*Fink et al., 2003*; *Li et al., 2008*). Since our data show that SMAD4-deficient CTLs maintained a consistent phenotype in the presence and absence of TGFβRII, we investigated whether R-SMAD2/3 were required to modify homing-receptor expression. Mice with flox sites in the genes for R-SMAD2 (*Ju et al., 2006*) and R-SMAD3 (*Li et al., 2008*) were used to generate CTLs that lack either R-SMAD2 (S2KO) or R-SMAD3 (S3KO), and both mutations (S23DKO). S3KO mice were further bred with S4KO mice to produce CTLs with both mutations (S34DKO). The respective Cre-deficient littermates were used as controls.

To study the effects of cytokine stimulation, naive CD8 T cells were isolated from SLO by negative selection and activated with plate-bound anti-CD3/CD28 plus recombinant IL-2 (20 u/ml). After 48 hr, the remaining CTLs were washed and cultured in fresh medium without T cell receptor (TcR) stimula-tion (48 hr). Replicate wells were supplemented with either rIL-2 alone or rIL-2 plus exogenous TGFβ (10 ng/ml). Selected wells were supplemented with an ALK5 inhibitor (SB431542) to block responses to any TGFβ in the media. At 96 hr after TcR stimulation, the cultured CTLs were analyzed for CD103 (*Figure 2A*) and CD62L expression (*Figure 2B*).

Naive CD8 T cells normally downregulate CD103 during antigen stimulation and expression returns when $T_{RM}$ cells are exposed to TGFβ (*Suarez-Ramirez et al., 2019*; *Teraki and Shiohara, 2002*). Consis-tently, the control CTLs did not express CD103 during culture with rIL-2 (plus and minus ALK5 inhib-itor), while this marker was induced during stimulation with TGFβ (*Figure 2A*). The same pattern of CD103 expression was observed when CTLs lacked R-SMAD2 (S2KO) or R-SMAD3 (S3KO). In contrast, S23DKO and TR2KO cells both lacked CD103 expression under all conditions analyzed, whereas S4KO, S4TR2KO, and S34KO cells expressed CD103 at intermediate levels. Sloan Kettering Institute (SKI) protein is a nuclear proto-oncogene that inhibits TGFβ signaling through interactions with SMAD proteins (*Sun et al., 1999*). A prior report found that SMAD4 interacts with SKI and creates heterodi-mers that repress *Itgae* expression (αE integrin/CD103) in CTLs (*Wu et al., 2021*). Consistently, S4KO cells expressed CD103 at higher levels than S4Ctrl cells during stimulation with rIL-2. However, CD103 expression increased when S4KO cells were stimulated with TGFβ. Conversely, CD103 expression did not change when S34KO and S4TR2-DKO cells were stimulated with TGFβ. These data show that R-SMAD2/3 are required for TGFβ to upregulate CD103 in the absence of SMAD4. When R-SMAD2 is absent, R-SMAD3 cooperates with SMAD4 to upregulate CD103 expression. Imaging flow cytometry confirmed that R-SMAD2/3 entered the nucleus without SMAD4 (*Figure 2—figure supplement 2*).

Our in vivo data show that low frequencies of OTI-S4KO cells reexpress CD62L during the memory phase of infection, indicating a defect during formation of $T_{CM}$ cells. Since TGFβ inhibits $T_{CM}$ formation (*Takai et al., 2013*), we analyzed the cultured CTLs for CD62L (*Figure 2B*). Between 50% and 60% of the control CTLs reexpressed CD62L when cultured with rIL-2 alone, and the percentages increased to approximately 100% when the ALK5 inhibitor was used. Conversely, only 25% of control CTLs expressed CD62L when cultured with TGFβ. Although 50% of S4KO, S34DKO, and S4TR2DKO cells expressed CD62L during culture with IL-2, the numbers did not change when the ALK5 inhibitor, or exogenous TGFβ, was used. TR2KO and S23KO cells expressed CD62L at high levels under all condi-tions analyzed. These data indicate that SMAD4 has dual functions during regulation of CD62L, since blocking TGFβ signaling was not sufficient to restore CD62L expression in the absence of SMAD4.

Because recombinant cytokines are not sufficient to induce KLRG1 expression on cultured CTLs (*Robbins et al., 2003*), we infected the mutant mice with LM-OVA. The bacteria were administered

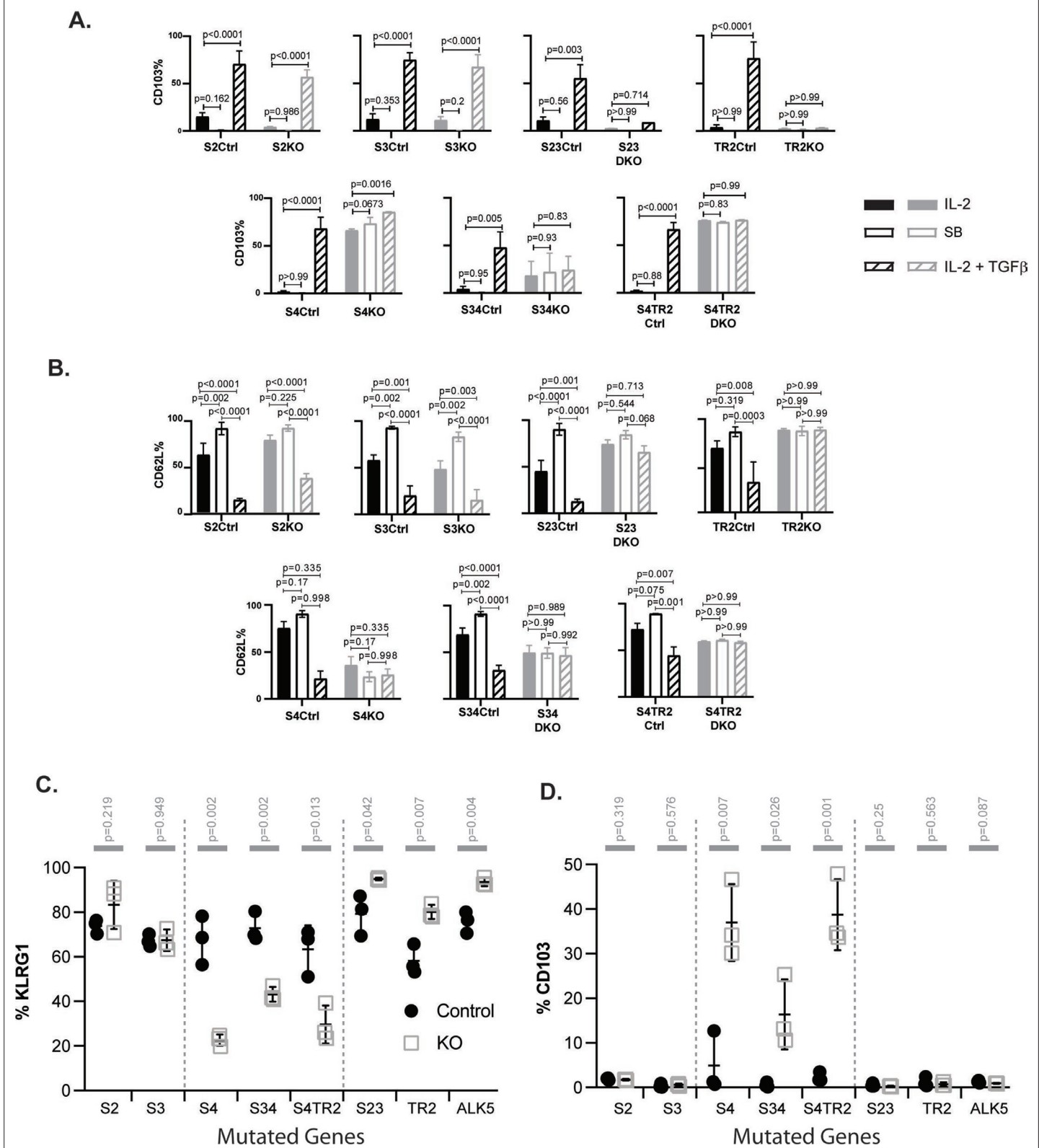

**Figure 2.** SMAD4 influences the fate decisions of pathogen-specific cytotoxic T lymphocytes (CTLs) independently of R-SMAD2/3. (**A, B**) Naive CD8 T cells from genetically modified mice were activated in vitro and stimulated with cytokines. Selected wells were supplemented with SB431542 (10 µM). Bars show CTLs stimulated with rIL-2 alone (no fill), rIL-2 plus SB431542 (gray fill), and rIL-2 plus transforming growth factor β (TGFβ) (hatched). Data are means ± standard deviation (SD) (3 mice per group). p values were calculated using Student's *t* tests. Two independent experiments produced similar

*Figure 2 continued on next page*

*Figure 2 continued*

results. (**A**) Percentages of CTLs that expressed CD103 at 96 hr after T cell receptor (TcR) stimulation. (**B**) Percentages of CTLs that expressed CD62L at 96 hr after TcR stimulation. (**C, D**) Different strains of genetically modified (empty squares) and control mice (filled circles) were infected (i.v.) with LM-OVA and at 8 dpi OVA-specific CTLs in the spleen were analyzed with MHCI tetramers. Data are means ± SD (n=3 per group). p values were calculated using two-way ANOVA and statistical comparisons were made using Tukey's multiple comparisons test. Two independent experiments produced similar results. (**C**) Scatter plot shows tetramer⁺ CTLs analyzed for KLRG1 expression. (**D**) Scatter plot shows tetramer⁺ CTLs analyzed for CD103 expression.

The online version of this article includes the following figure supplement(s) for figure 2:

**Figure supplement 1.** C57BL/6 mice were infected with X31-OVA T_RM and T_CM cells were isolated from the lungs and spleens at 60 dpi.

**Figure supplement 2.** OTI-S4Ctrl and OTI-S4KO were transferred to C57BL/6 mice before infection with LM-OVA.

by i.v. injection and MHCI tetramers were used to analyze OVA-specific CTLs in the spleens at 8 dpi (*Figure 2C, D*). Between 50% and 90% of the control CTLs expressed KLRG1 and the percentages did not change in the absence of either R-SMAD2 (S2KO) or R-SMAD3 (S3KO). Large percentages of CTLs expressed KLRG1 (80–90%) in the absence of R-SMAD2/3 (S23DKO), or a functional TGFβ receptor (TR2KO and ALK5KO), confirming that expression was repressed by TGFβ (*Sanjabi et al., 2009*). Conversely in the absence of SMAD4 (S4KO, S34DKO, and S4TR2DKO) only small percentages of CTLs expressed KLRG1, while the portion of CD103 expressing cells significantly increased (*Figure 2D*). The phenotypic changes in S34DKO cells were slightly less pronounced. The reason for this is unclear, but may reflect the influence of a feedback mechanism. Our data show that SMAD4 promotes formation of terminally differentiated T_EFF cells via a mechanism that does not require TGFβ, or R-SMAD2/3.

## The transcriptional profile of SMAD4-deficient CTLs is similar to T_RM cells

TGFβ is an important regulatory factor during CTL development, but relatively little is known about the function of SMAD4 during transcriptional programming of activated CTLs. To address this question, we used RNA-sequencing to compare the transcriptional profiles of SMAD4-deficient and control CTLs during IAV infection. Early effector (EE) cells are partially differentiated CTLs that have not committed to a classical SLEC/MPEC phenotype, and can be identified using high CD44/CD11a expression in the absence of KLRG1, CD127, and CD103 (*Plumlee et al., 2015*). To compare similar populations of antigen-specific CTLs, naive OTI-S4KO and OTI-S4Ctrl cells were transferred to B6 mice before infection with X31-OVA and EE cells were isolated from the spleens at 6 dpi. Samples were analyzed in triplicate and mRNA-sequences were registered with the GEO repository (accession # GSE151637). Differentially expressed genes were ranked according to fold change in OTI-S4KO cells (*Figure 3A* and *Figure 3—source data 1*). We found that OTI-S4KO cells expressed EOMES (*Eomes*) and CD62L (*Sell*) at lower levels than the controls, while *Itgae* was upregulated (*Figure 3B*). Since CD103 is expressed on large numbers of T_RM cells in mucosal tissues, we used gene set enrichment analysis (GSEA) to compare the transcriptional profile of OTI-S4KO cells with previously published datasets. The GSEA plots show comparisons with T_RM cells in the small intestine (*Milner et al., 2017*; *Figure 3C*) and other tissues (*Figure 3D*). Many signature genes that are associated with T_RM cells in the small intestine were upregulated in OTI-S4KO cells. We found similar correlations with T_RM cells isolated from the brain and liver, as well as CTLs that were stimulated with TGFβ in vitro (*Mackay et al., 2016*; *Milner et al., 2017*; *Nath et al., 2019*; *Wakim et al., 2012*). These correlations indicate that early signaling via SMAD4 impedes differentiation toward a resident memory phenotype.

We used qPCR to further confirm that SMAD4 altered *Itgae* expression in vitro. Naive CD8 T cells were activated with plate-bound anti-CD3/CD28 plus recombinant IL-2 (20 u/ml). After 48 hr, RNA was extracted from S4Ctrl cells to establish a baseline for gene expression. The remaining CTLs were cultured in fresh medium without TcR stimulation. Replicate wells were supplemented with either rIL-2 alone or rIL-2 plus exogenous TGFβ (10 ng/ml). RNA was extracted at 24 and 48 hr after cytokine stimulation (i.e., 72 and 96 hr after TcR stimulation, respectively) for qPCR analysis (*Figure 3E*). The graphs show relative changes in *Itgae* expression (log₂ fold change) compared to the controls. As expected, the S4KO cells produced more *Itgae* transcripts than S4Ctrl cells during culture with IL-2 and expression increased when TGFβ was added.

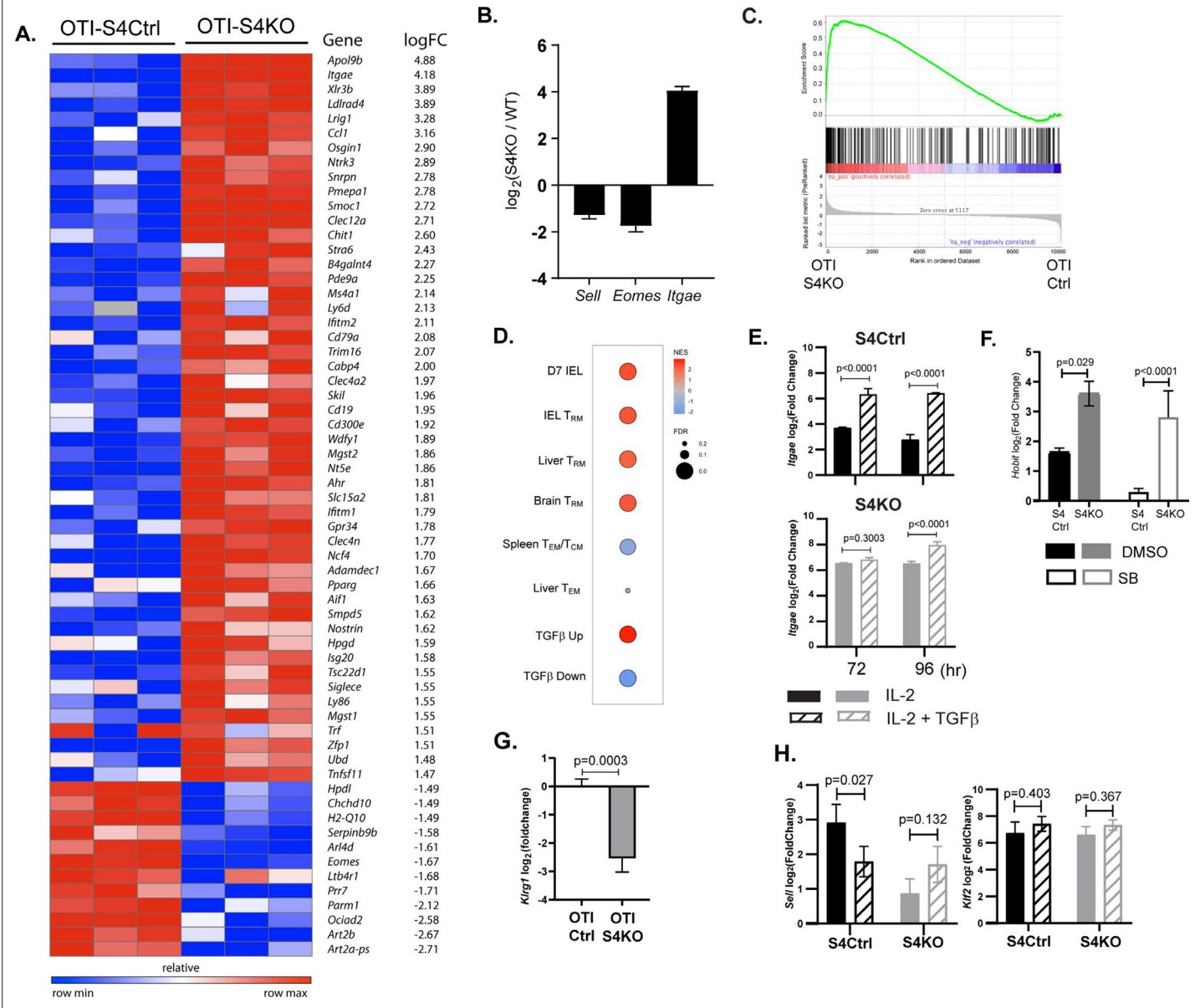

**Figure 3.** The transcriptional profile of SMAD4-deficient cytotoxic T lymphocytes (CTLs) resembles a $T_{RM}$ phenotype. (**A–C**) OTI-S4Ctrl and OTI-S4KO were transferred to B6 mice before infection with X31-OVA. Early effector (EE) cells were analyzed 6 dpi, using RNA isolated from pools of three spleens. Samples were analyzed in triplicate (total n = 9/group). (**A**) Differentially expressed genes are ranked according to fold change in OTI-S4KO cells. (**B**) OTI-S4KO cells expressed *Eomes* and *Sell* at reduced levels while *Itgae* was induced. (**C, D**) The transcriptional profile of OTI-S4KO cells was compared with published datasets. (**C**) Comparison with IEL analyzed 7 dpi (GSE107395). (**D**) Comparisons with $T_{RM}$ cells in brain (GSE39152), $T_{RM}$, and $T_{EM}$ cells in liver (GSE70813), $T_{CM}$ cells in spleen (GSE70813), and in vitro activated CTLs (GSE125471). Color scale indicates normalized enrichment scores (NESs) and circles indicate false discovery rates (FDRs). (**E**) S4Ctrl (top graph) and S4KO cells (bottom graph) were analyzed for *Itgae* transcripts at 72 and 96 hr after T cell receptor (TcR) stimulation. Bars are cells stimulated with rIL-2 alone (filled bars) and rIL-2 plus transforming growth factor β (TGFβ) (hatched bars). Samples were analyzed in triplicate. Two independent experiments gave similar results. (**F**) Cultured CTLs were analyzed for *Hobit* transcripts at 96 hr after TcR stimulation. Bars show cells stimulated with rIL-2 alone (filled bars), and rIL-2 plus ALK5 inhibitor (white bars). Samples were analyzed in triplicate. Two independent experiments gave similar results. (**G**) qPCR was used to measure *Klrg1* transcripts in OTI-S4KO and OTI-S4Ctrl cells at 8dpi with LM-OVA. Samples were analyzed in triplicate. Two independent experiments gave similar results. (**H**) Cultured CTLs were analyzed for *Sell* transcripts at 96 hr after TcR stimulation. Bars show CTLs stimulated with rIL-2 alone (filled bars) and rIL-2 plus TGFβ (hatched bars). Samples were analyzed in triplicate. Two independent experiments gave similar results. p values were calculated using Student's *t* tests.

The online version of this article includes the following source data for figure 3:

**Source data 1.** Source data for RNA-sequencing from EE cells (OTI-S4KO and OTI-S4Ctrl) analyzed 6 dpi with X31-OVA.

Many transcription factors have important functions during CD8 T cell differentiation. Notably, homolog of Blimp1 (Hobit) promotes formation of $T_{RM}$ cells (*Mackay et al., 2016*; *Parga-Vidal et al., 2021*) and EOMES is expressed in $T_{CM}$ cells (*Intlekofer et al., 2005*) and downregulated by TGFβ as $T_{RM}$ cells settle in peripheral tissues (*Mackay et al., 2015*). Since the sequencing data indicate that SMAD4-deficient CTLs are transcriptionally related to $T_{RM}$ cells, we used qPCR to measure *Hobit* transcripts in vitro. Naive S4KO and S4Ctrl cells were activated with antibodies (48 hr) and cultured with rIL-2 (48 hr) in the presence and absence of the ALK5 inhibitor. Samples were analyzed by qPCR at 96 hr after TcR stimulation (*Figure 3F*). The S4KO cells produced more *Hobit* transcripts than S4Ctrl cells during culture with IL-2. The numbers of *Hobit* transcripts in the S4Ctrl cells decreased when the ALK5 inhibitor was used, while expression in S4KO cells did not change. These data indicate that *Hobit* is induced by TGFβ and downregulated via SMAD4.

Because neuraminidase activates TGFβ, small percentages of OTI-Ctrl cells expressed KLRG1 during infection with X31-OVA (*Figure 1B*). Consequently, we did not detect significant differences in the numbers of *Klrg1* transcripts in OTI-S4KO and OTI-S4Ctrl cells by RNA-sequencing. To generate large populations of KLRG1[+] CTLs, we transferred OTI-S4KO and OTI-S4Ctrl cells to B6 mice before infection with LM-OVA and analyzed donor cells in the spleen at 8 dpi using qPCR (*Figure 3G*). The data show that OTI-S4KO cells produced fewer *Klrg1* transcripts than OTI-S4Ctrl cells during the $T_{EFF}$ response.

L-selectin (CD62L) is required for naive and $T_{CM}$ cells to enter resting lymph nodes via HEV. Krupple-like factor 2 (KLF2) plays a central role in transcriptional regulation of CD62L during the CTL response (*Takada et al., 2011*). Since our data suggest that CD62L is regulated via SMAD4 (*Cao et al., 2015*; *Hu et al., 2015*), we measured *Sell* (CD62L) and *Klf2* transcripts in cultured CTLs after cytokine stimulation (*Figure 3H*). Naive S4KO and S4Ctrl cells were activated (48 hr) and stimulated with rIL-2, or rIL-2 plus TGFβ (48 hr) as previously and RNA was extracted for qPCR analysis. S4KO cells produced fewer *Sell* transcripts than S4Ctrl cells during culture with rIL-2. The numbers of *Sell* transcripts in S4Ctrl cells decreased during stimulation TGFβ, while expression in S4KO cells did not change. These data indicate that CD62L is induced via SMAD4, and downregulated by TGFβ. The numbers of *Klf2* transcripts were not modified by SMAD4 or TGFβ.

## EOMES is cooperatively regulated by TGFβ and SMAD4

EOMES plays a major role in lineage specification of pathogen-specific CTLs. Our sequencing data show that OTI-S4KO cells produce fewer *Eomes* transcripts than OTI-S4Ctrl cells during IAV infection (*Figure 3B*). To confirm whether EOMES was induced via SMAD4, we activated S4KO and S4Ctrl cells in vitro (48 hr) and measured *Eomes* transcripts by qPCR (*Figure 4A*). The numbers of *Eomes* transcripts in S4Ctrl cells transiently decreased during culture with rIL-2 (72 hr after TcR stimulation) and returned to baseline at 96 hr. *Eomes* was downregulated by TGFβ at both time points. Importantly, S4KO cells produced very few *Eomes* transcripts that S4Ctrl cells during culture with IL-2 and did not respond to TGFβ. These data confirmed that EOMES was induced via SMAD4 and downregulated by TGFβ.

We next used transfer experiments to monitor changes in EOMES expression in vivo. Naive OTI-S4KO, OTI-S4Ctrl, OTI-TR2KO, and OTI-TR2Ctrl mice were transferred to B6 mice before infection with LM-OVA. On different dpi, donor cells in the spleens were analyzed for EOMES expression by intracellular staining (*Figure 4B, C*). OTI-TR2KO cells expressed EOMES at higher levels than OTI-TR2Ctrl cells at all time point analyzed (*Figure 4B*), whereas OTI-S4KO cells expressed EOMES at lower levels than OTI-S4Ctrl cells (*Figure 4C*). After infection with X31-OVA, the differences in EOMES expression were less pronounced, which may reflect the influence of virally induced TGFβ (*Figure 4—figure supplement 1*). These EOMES are induced via SMAD4 and downregulated by TGFβ.

When CTLs were analyzed after LCMV infection, imaging flow cytometry showed that EOMES was concentrated in the nuclei of exhausted CTLs (*McLane et al., 2021*). We used imaging flow cytometry to determine whether EOMES translocated into the nuclei of OTI-S4KO cells during $T_{EFF}$ formation (*Figure 4D-F*). Donor cells were transferred to B6 mice before infection with LM-OVA and the spleens were analyzed at 8 dpi. The OTI-S4KO cells expressed EOMES at lower levels than OTI-S4Ctrl, and a smaller percentage of the protein was located in the nuclei where it could influence gene expression (*Figure 4E*). We analyzed TR2KO and TR2Ctrl cells for EOMES expression using a KLRG1 gate (*Figure 4F*). KLRG1[+] CTLs from TR2KO and TR2Ctrl mice show similar patterns of EOMES expression.

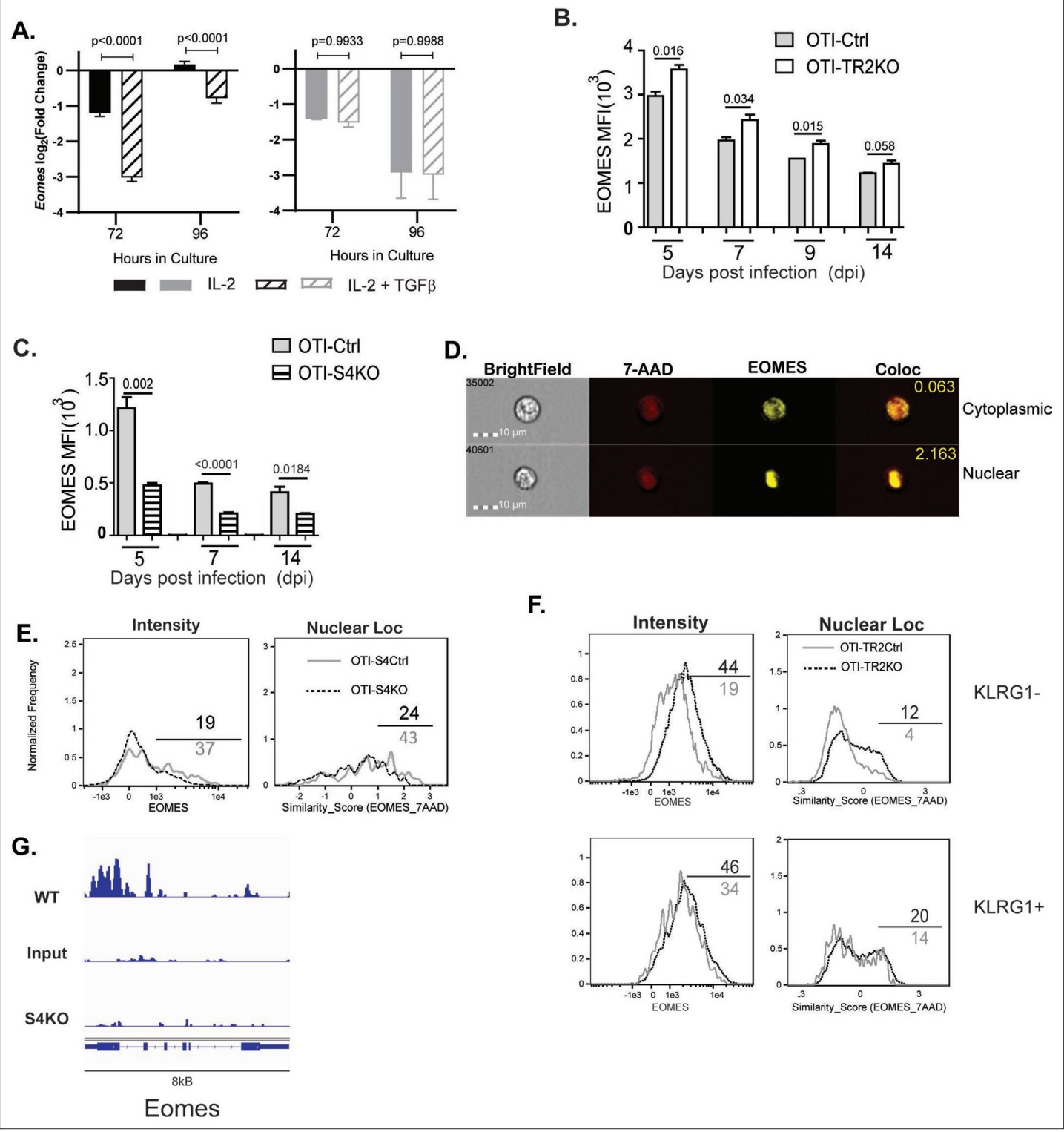

**Figure 4.** EOMES expression is induced via SMAD4 and downregulated by transforming growth factor β (TGFβ). (**A**) S4Ctrl (left) and S4KO (right) cells were analyzed for *Eomes* transcripts at 96 hr after T cell receptor (TcR) stimulation. Bars show cytotoxic T lymphocytes (CTLs) stimulated with rIL-2 alone (filled bars) and rIL-2 plus TGFβ (hatched bars). Samples were analyzed in triplicate. Two independent experiments gave similar results. p values were calculated using Student's *t* tests. (**B, C**) OTI-S4KO, OTI-S4Ctrl, OTI-S4TR2TKO, and OTI-S4TR2Ctrl cells were transferred to B6 mice before infection with LM-OVA. On different days after infection, donor cells in the spleens were analyzed for EOMES expression by intracellular staining. Data are mean ± standard deviation (SD) (*n* = 4/group). Two independent experiments gave similar results. p values were calculated using Student's *t* tests. (**B**) EOMES

*Figure 4 continued on next page*

*Figure 4 continued*

expression (MFI) in OTI-TR2KO (no fill) and OTI-TR2Ctrl (gray fill). (**C**) EOMES expression (MFI) in OTI-S4KO (hatched) and OTI-S4Ctrl (gray fill). (**D–F**) Donor cell were transferred to B6 mice before i.n. infection with LM-OVA and analyzed by imaging flow cytometry at 8 dpi. Data are means ± SD, *n* = 3. (**D**) Representative staining for EOMES (yellow) and 7-AAD (red). (**E**) Histograms show EOMES intensity and similarity scores for OTI-S4Ctrl (gray line) and OTI-S4KO (dashed line) cells. (**F**) Histograms show EOMES intensity and similarity scores for OTI-TR2Ctrl (gray line) and OTI-TR2KO (dashed line) cells. (**G**) Published ChIP-Seq data from in vitro activated CTLs (GSE135533) were used to identify SMAD4-binding sites in the EOMES locus (8 kb).

The online version of this article includes the following figure supplement(s) for figure 4:

**Figure supplement 1.** Naïve CD8 T cells were transferred to C57BL/6 mice before infection with X31-OVA.

Gating on KLRG1$^{-negative}$ cells showed that TR2KO cells expressed EOMES at higher levels than the controls (*Figure 4F*). EOMES expression in KLRG1+ CTLs was not altered by TGFb. Since our data showed that EOMES was induced via SMAD4, we used published ChIP-Seq data (*Wu et al., 2021*) to determine whether the regulatory sequences in the *Eomes* gene include SMAD4-binding sites (*Figure 4*). The data show that the promoter region for the *Eomes* gene includes multiple SMAD4-binding sites.

## Ectopic EOMES expression alters the ratios of CTLs that express CD103 and CD62L

Prior studies indicate that EOMES must be downregulated before T$_{RM}$ cells express CD103 (*Mackay et al., 2015*). Since our data showed that OTI-S4KO cells consistently expressed EOMES at reduced levels, we reasoned that SMAD4 might indirectly influence homing-receptor expression by altering EOMES expression. To explore this possibility, we intercrossed S4KO and EOMES-deficient (EKO) mice to produce CTLs with both mutations (ES4KO). Cre-deficient littermates (ECtrl, S4Ctrl, and ES4Ctrl) were used as controls. Naïve CD8 T cells were activated in vitro (48 hr) and stimulated with cytokines (48 hr) as previously. Selected wells were supplemented with the ALK5 inhibitor to prevent signaling via the TGFβ receptor. CTLs were analyzed for CD103 (*Figure 5A*) and CD62L expression (*Figure 5B*) at 96 hr after TcR stimulation.

As expected, the control cells did not express CD103 during culture with IL-2 and this marker was induced by TGFβ. The S4KO and EKO cells both expressed CD103 at intermediate levels during culture with rIL-2 and expression increased when TGFβ was added. Conversely, ES4KO cells expressed CD103 at high levels under all conditions analyzed. These data show that the *Itgae* gene was repressed by two different mechanisms. In wildtype mice, CD62L is present on naïve CD8 T cells, then downregulated during T$_{EFF}$ formation and reexpressed on T$_{CM}$ cells. We analyzed naïve CD8 T cells before activation and found similar levels of CD62L expression in each group of mice (*Figure 5—figure supplement 1*). The control CTLs downregulated CD62L at 48 hr after TcR stimulation in vitro (*Figure 5—figure supplement 2*), while a majority CTLs reexpressed CD62L during culture with rIL-2. When cultured cells were stimulated with cytokines, the percentages of control CTLs that expressed CD62L increased when the ALK5 inhibitor was present and decreased when TGFβ was used. In contrast, only 50% of S4KO and ES4KO cells expressed CD62L under all conditions analyzed (*Figure 5B*). These data confirmed that antigen-experienced CTLs downregulate CD62L in response to TGFβ, and that SMAD4 is required for CTLs to reexpress CD62L during memory formation.

To compare homing-receptor expression in vivo, we infected S4KO, EKO, and ES4KO mice (plus littermate controls) with X31-OVA and analyzed CTLs at 30 dpi (*Figure 5C*). Anti-CD8 antibodies were given by i.v. injection 3 min before sacrifice. OVA-specific CTLs were analyzed using MHCI tetramers. The lungs and draining lymph nodes (MLN) from each group of mice contained some CD103$^+$ T$_{RM}$ cells. Abnormal CD103 expression was detected on OVA-specific CTLs in the spleens of S4KO and ES4KO mice. Approximately 50% of OVA-specific CTLs in the ILNs from control mice expressed CD62L, indicating the presence of T$_{CM}$ cells. Smaller percentages of OVA-specific CTLs expressed CD62L in the ILNs of S4KO, EKO, and ES4KO mice (17–22%), supporting the idea that signaling via SMAD4 promotes formation of T$_{CM}$ cells.

We next explored whether ectopic EOMES expression altered the phenotype of S4KO cells. A murine stem cell virus (MSCV) vector that encodes green fluorescent protein (MSCV)-IRES-GFP (pMIG) was used to overexpress EOMES in antigen-experienced CTLs in vitro (*Figure 5D*) and in vivo (*Figure 5E*). The pMIG vector has been described previously (*Pearce et al., 2003*). Naïve S4KO and EKO cells were stimulated with anti-CD3/28 and rIL-2 (48 hr), then transduced with either

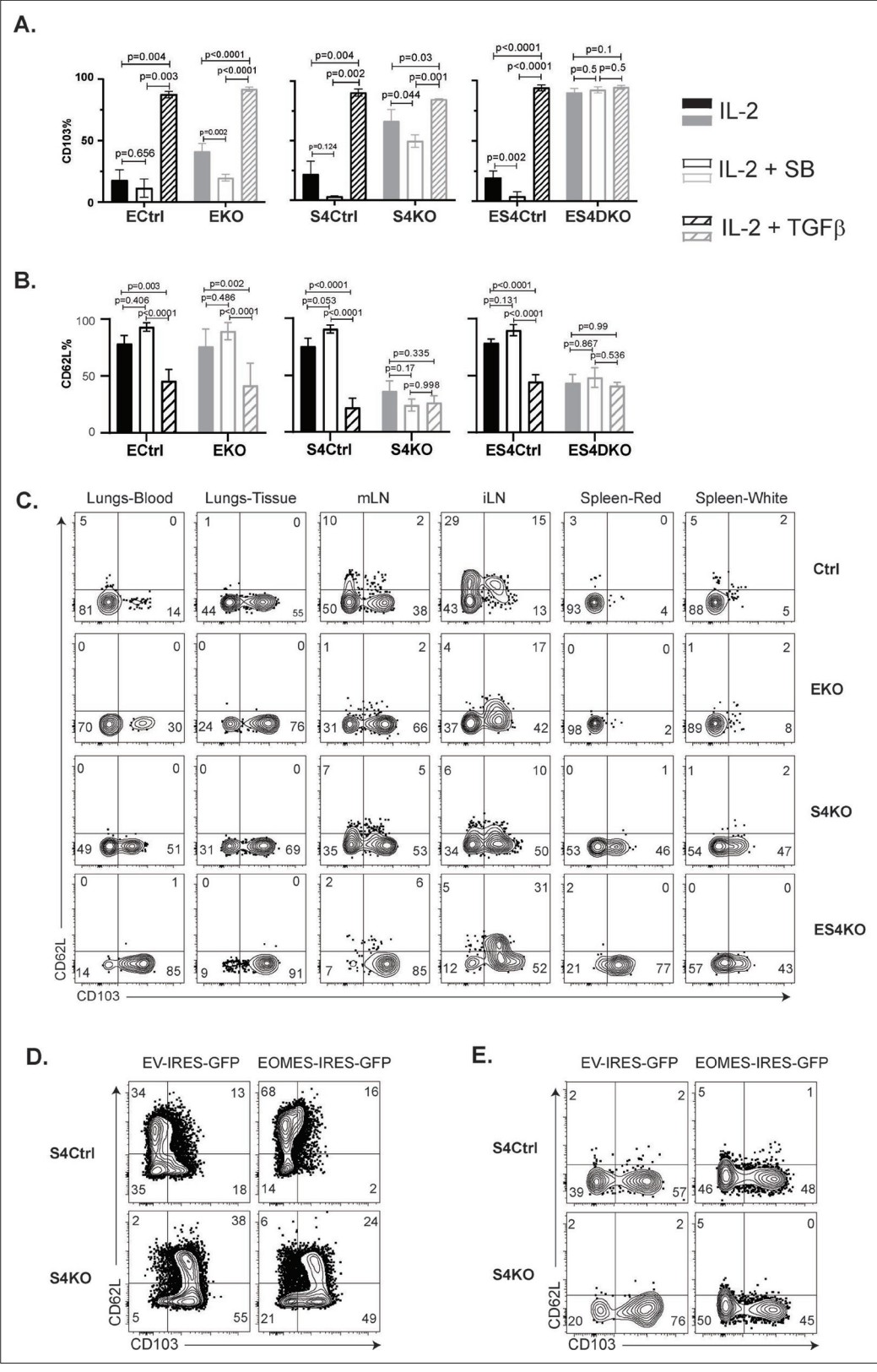

**Figure 5.** Regulation of EOMES via SMAD4 alters homing-receptor expression. (**A, B**) S4KO, EKO, ES4KO
(gray), and control mice (black) were activated in vitro and stimulated with cytokines. Graphs show cytotoxic
T lymphocytes (CTLs) analyzed 96 hr after T cell receptor (TcR) stimulation. Bars are CTLs stimulated with rIL-2
alone (filled bars), rIL-2 plus SB431542 (no fill), and rIL-2 plus transforming growth factor β (TGFβ) (hatched). Data

*Figure 5 continued on next page*

*Figure 5 continued*

are means ± standard deviation (SD) (3 mice per group). p values were calculated using Student's *t* tests. Two independent experiments produced similar results. (**A**) Percentages of CD8 T cells that expressed CD103. (**B**) Percentages of CD8 T cells that expressed CD62L. (**C**) S4KO, EKO, and ES4KO mice were infected with X31-OVA and anti-viral CTLs were analyzed with MHCI tetramers at 33 dpi. Anti-CD8 antibodies were injected 5 min before sacrifice. Contour plots show OVA-specific CTLs analyzed for CD103 and CD62L expression. (**D, E**) S4KO and S4Ctrl cells were activated in vitro (48 hr) and transduced with a retroviral vector encoding EOMES (Eomes-IRES-GFP) or the empty vector control (EV-IRES-GFP). (**D**) GFP⁺ cells were analyzed for CD103 and CD62L expression after 6 days in culture. (**E**) S4KO and S4Ctrl cells were transferred to B6 mice that were previously infected (24 hr) with X31-OVA. GFP⁺ cells in the lungs were analyzed for CD62L and CD103 expression at 35 dpi. Data are means ± SD (3 mice per group). Two independent experiments produced similar results.

The online version of this article includes the following figure supplement(s) for figure 5:

**Figure supplement 1.** Naïve CD8 T cells from gentically modified mice were analyzed for CD62L expression by flow cytometry.

**Figure supplement 2.** Naïve CD8 T cells from the spleens of C57BL/6 mice were stimulated with anti-CD3/28 (plus IL-2).

**Figure supplement 3.** EKO cells were transduced with EOMES-IRES-GFP (dashed lines) or EV-IRES-GFP (gray fill).

**Figure supplement 4.** S4Ctrl and S4KO CD8 T cells were transduced with EV-IRES-GFP or EOMES-IRES-GFP.

**Figure supplement 5.** S4KO CD8 T cells were transduced with EV-IRES-GFP or SMAD4-IRES-GFP.

EOMES-IRES-GFP, or empty vector control (EV-IRES-GFP). At 96 hr after TcR stimulation, the GFP⁺ CTLs were analyzed for CD103 expression by flow cytometry. EKO cells were used to verify that the RV-EOMES-GFP vector was functional (overlaid histograms) (*Figure 5—figure supplement 3*). When EOMES was expressed ectopically, smaller percentages of S4KO and S4Ctrl cells expressed CD103, while the percentages of CD62L⁺ CTLs increased (*Figure 5D* and *Figure 5—figure supplement 4*). GFP⁺ cells in lungs expressed CD103 at reduced levels after transduction with the EOMES-IRES-GFP vector (*Figure 5E*). These experiments show that ectopic EOMES expression partially restored the phenotype of S4KO cells, and induced a shift toward a central memory phenotype. We also used the (MSCV)-IRES-GFP (pMIG) vector to express SMAD4 ectopically and also observed a shift toward a T$_{CM}$ phenotype in S4KO cells (*Figure 5—figure supplement 5*).

## Discussion

Different approaches have been used to determine how specialized subsets of pathogen-specific CTLs acquire heritable traits. Several lines of evidence indicate that naïve CD8 T cells are not a uniform cell population. Early studies used limiting-dilution assays and fate-mapping experiments to show that naïve CD8 T cells give rise to mixed populations of T$_{EFF}$ and memory CD8 T cells (*Plumlee et al., 2015*; *Stemberger et al., 2007*). Others found unequal inheritance of genetic markers during T$_{EFF}$ formation, indicating that T$_{EFF}$ and memory CD8 T cells are products of asymmetric cell division (*Chang et al., 2011*; *Pollizzi et al., 2016*). Other studies indicate that the differentiation program for memory CD8 T cells begins soon after antigen stimulation, as the IL-7 receptor (CD127) is expressed CTLs that are predisposed toward memory formation and absent from T$_{EFF}$ cells that express KLRG1 (*Kaech et al., 2003*). Transcriptome analysis also revealed the presence of T$_{RM}$ precursors in a pool of circulating T$_{EFF}$ cells (*Kok et al., 2020*). Other data indicate that programming occurs at an early stage of diffrentiation, since migratory DCs that activate TGFβ can precondition naïve CD8 T cells to become T$_{RM}$ cells that localize in peripheral tissues (*Mani et al., 2019*). Although many studies support diverging differentiation pathways, the question of whether specialized traits are genetically programmed, or induced by external stimuli has not been resolved. In an attempt to address this question, investigators used transfer experiments to track clonal populations of antigen-specific CD8 T cells during infections with different type of pathogens. The study showed that the population dynamics reflected the inflammatory environment that developed during infection (*Plumlee et al., 2013*). Large numbers of T$_{EFF}$ cells expressed KLRG1 during infection with LM, while this marker was less prevalent during infection with *Vesicular Stomatitis Virus*. Our study builds on this observation by revealing the requirement for SMAD-dependent sugnling pathways during fate-determination of pathogen-specific CTLs.

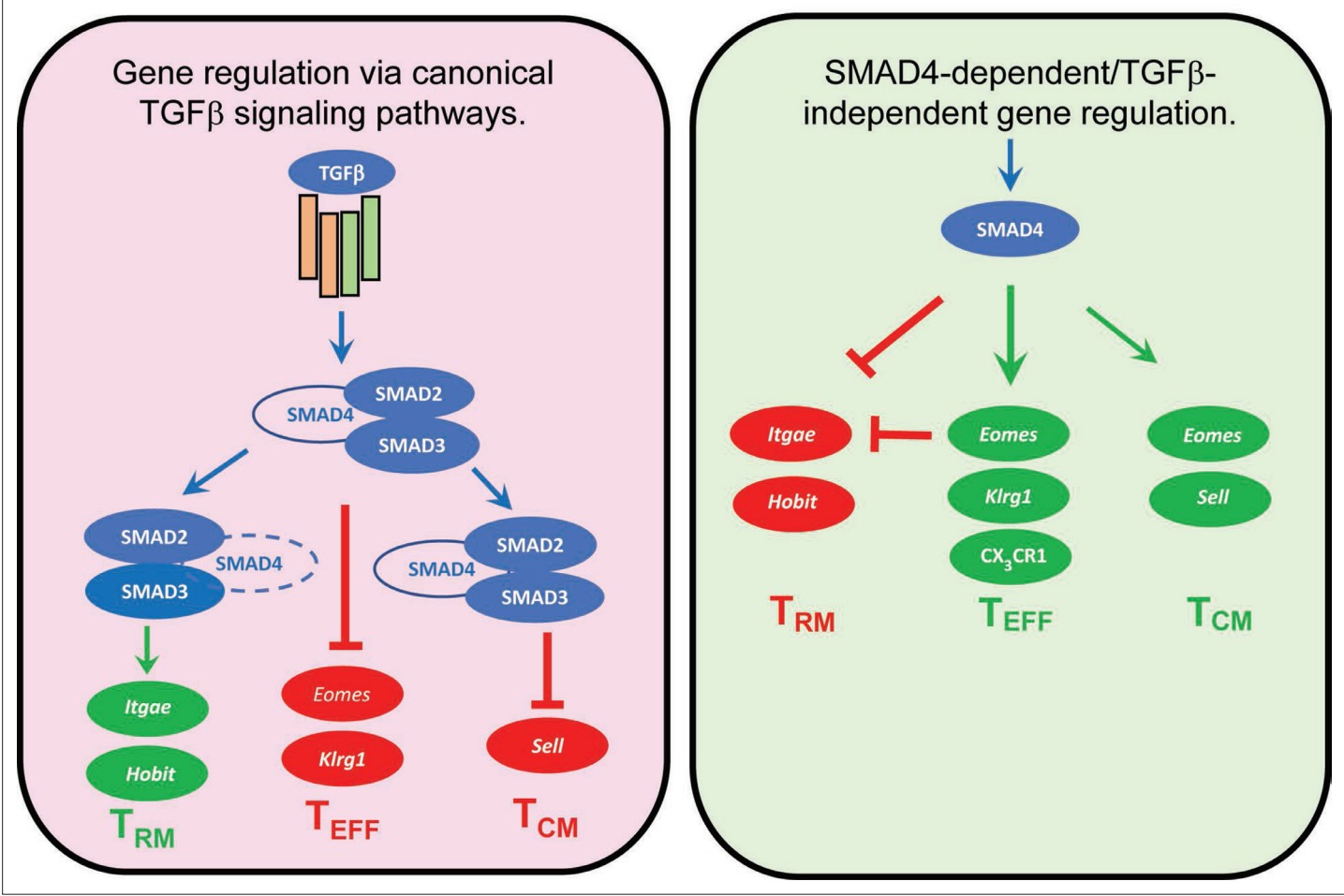

**Figure 6.** Genes that are cooperatively regulated by SMAD4 and transforming growth factor β (TGFβ). Diagram shows reciprocal changes in gene expression as CTLs are modified via SMAD-dependent signaling pathways.

Several cytokines are known to influence the 'fate decisions' of newly activated CTLs, including IL-12 which supports $T_{EFF}$ formation (*Mescher et al., 2006*), whereas TGFβ has anti-inflammatory properties and encourages $T_{RM}$ cells to settle in peripheral tissues (*Skon et al., 2013*). Several studies show that signaling via SMAD4 is required for formation of $T_{EFF}$ and $T_{CM}$ cells, whereas TGFβ creates a bias toward $T_{RM}$ development (*Cao et al., 2015*; *Hu et al., 2015*; *Mackay et al., 2015*). Our current study extends this work by showing that SMAD4 and TGFβ are essential components of two distinct signaling pathways that guide the fate decisions of newly activated CTLs using alternative genetic programs. Importantly, these opposing pathways control a key bifurcation in the differentiation pathway of antiviral CTLs, by regulating the same genes in opposite directions (*Figure 6*). We show that SMAD4 upregulates genes that are highly expressed in CTLs in the circulation, including EOMES, KLRG1, and CD62L. The same genes are downregulated by TGFβ as $T_{RM}$ cells settle in peripheral tissues (*Mackay et al., 2015*; *Sanjabi et al., 2009*). There is evidence that cell-state impacts susceptibility to TGFβ, since we previously found that $T_{CM}$ cells did not upregulate CD103 during stimulation with TGFβ (*Suarez-Ramirez et al., 2019*) and our current study shows that KLRG1⁺ CTLs maintain EOMES expression in the presence of TGFβ (*Figure 4F*). Although studies show that SMAD4 acts independently of TGFβ to support commitment to $T_{EFF}$ and $T_{CM}$ lineages, further work is required to determine whether SMAD4-dependent epigenetic modifications are required for CTLs to become terminally differentiated.

SMAD proteins modulate gene expression through interactions with numerous transcription factors, including the protooncogene SKI and forms heterodimers that repress the *Itgae* gene (CD103) (*Wu et al., 2021*). Our data show that CD103 is also downregulated via SMAD4-dependent induction

of EOMES. Others used retroviral vectors to ectopically express T-box transcription factors in activated CTLs and found reduced numbers of $T_{RM}$ cells in the skin of HSV infected mice when EOMES and T-bet were expressed at high levels (*Mackay et al., 2015*). Consistently, others found that reduced EOMES and T-bet expression facilitated $T_{RM}$ development (*Laidlaw et al., 2014*). To further understand how these transcription factors support memory formation, we investigated whether EOMES was regulated via SMAD4. Our experiments confirmed that EOMES was induced via SMAD4 and downregulated by canonical TGFβ signaling. ChIP-Seq data show that there are multiple SMAD4-binding sites in the *Eomes* loci.

Many transcription factors have important functions during CD8 T cell differentiation. The major players include Zeb1 and Zeb2, which are homologous proteins that serve opposing functions during memory formation and are inversely regulated by TGFβ. *Zeb1* expression increases in maturing memory CD8+ T cells during exposure to TGFβ, whereas *Zeb2* is downregulated (*Dominguez et al., 2015*; *Guan et al., 2018*). Other studies indicate that DNA-binding inhibitors Id2 and Id3 also have reciprocal functions during fate determination of activated CTLs (*Ji et al., 2011*; *Omilusik et al., 2018*; *Yang et al., 2011*). Whether SMAD4 alters the expression levels of these transcription factors, or other epigenetic regulators, during CD8 T cell differentiation needs to be further investigated.

## Materials and methods
### Mice and reagents
Mice were bred and housed at the University of Connecticut Health Center in accordance with institutional guidelines. Experiments were performed in accordance with protocols approved by the UCONN Health Institutional Animal Care and Use Committee (IACUC). Mice that express Cre-recombinase under the control of the distal-Lck promoter (dLck) were used to generate mice that lack R-SMAD2 (S2KO), R-SMAD3 (S3KO), co-SMAD4 (S4KO), EOMES (EKO), and TGFβ receptor II (TR2KO). S4KO and TR2KO were further crossed with OTI mice that express a transgenic antigen receptor specific for the SIINFEKL peptide presented on H-2K$^b$.

Virus stocks were grown in fertilized chicken eggs (Charles River) and stored as described previously. Between 8 and 20 weeks after birth, anesthetized mice were infected intranasally (i.n.) with either $2 \times 10^3$ plaque-forming units (PFU) X31-OVA, or $5 \times 10^3$ colony-forming units (CFU) of recombinant *L. monocytogenes* expressing chicken ovalbumin (LM-OVA).

### Sample preparation for adoptive transfer and flow cytometry
Naive CD8 T cells were isolated from SLO using Mojosort isolation kits (Biolegend, Dedham MA). Mice received $5 \times 10^3$ congenically marked donor cells by intravenous (i.v.) injection given 48 hr before infection. To identify bloodborne CTLs by IV staining, mice were injected with 1 µg anti-CD8β in 200 µl PBS and sacrificed 5 min later. For flow cytometry, chopped lung tissue was incubated at 37°C for 90 min in RPMI with 5% fetal bovine serum (FBS) and 150 U/ml collagenase (Life Technologies, Rockville, MD, USA). Nonadherent cells were enriched on Percoll density gradients (44/67%) spun at 1200 × *g* for 20 min. Lymphocytes were incubated with antibodies that block Fc-receptors (15 min at RT). Antigen-experienced CD8 T cells were identified using high CD11a/CD44 expression and analyzed with antibodies specific for CD103, KLRG1, and CD62L. For intracellular staining, lymphocytes were analyzed using True Nuclear transcription factor buffer (Biolegend, Dedham, MA). Permeabilized cells were stained with antibodies specific for T-bet and EOMES. To monitor cell proliferation, cells were labeled with CFSE dye before transfer to C57BL/6 mice. Alternatively, mice received a single injection of BrdU by i.p. injection at 30 dpi and donor cells were analyzed 3–6 hr later, using BrdU analysis Kits (BD Pharmingen). Stained cells were analyzed using a BD LSRII with FACSDiva V8.0 (BD Biosciences) and processed using Flowjo and GraphPad Prism software. To visualize intracellular proteins by imaging cytometry, fixed cells were permeabilized using 90% ice-cold methanol for intracellular staining. CTLs were imaged at ×60 normal magnification using an Amnis ImagestreamX MKII flow cytometer. Similarity scores were calculated using Amnis Ideas V6.2 (Luminex) software. Higher similarity scores indicate protein translocation into the nucleus.

## Cell culture

Naive CD8 T cells were grown in RPMI with 10% FBS, L-glutamine, β-mercapthoethanol, sodium pyruvate, HEPES (4-(2-hydroxyethyl)-1-piperazineethanesulfonic acid), and antibiotics. For TcR stimulation, CTLs were stimulated with plate-bound anti-CD3/CD28 (1 μg/ml) and rIL-2 (20 U/ml). After 48 hr, CTLs were transferred to new wells for cytokine stimulation without TcR stimulation. Replicate wells were supplemented with rIL-2 (20 U/ml), ALK5 inhibitor (SB431542, 10 μM/ml), and TGFβ (10 ng/ml).

## RNA-sequencing and GSEA

Messenger-RNA was extracted from purified CTLs using RNAeasy Plus Mini Kits (Qiagen, Hilden Germany) and sequenced by Otogenetics Corporation (Atlanta, GA). RNA-sequences from OTI-S4KO and OTI-Ctrl cells are available in the GEO database (accession # GSE151637). Data were analyzed using Tophat2, Cufflinks, and Cuffdiff software (DNAnexus, Mountain View, CA). Transcriptional changes greater than twofold (Log2 fold change >1 or <−1, p < 0.05, false discovery rate <0.05) were used to select genes for Ingenuity pathway (Qiagen, Hilden Germany) and GSEA (Broad Institute, San Diego, CA) (*Subramanian et al., 2005*). Differentially expressed genes were compared with published datasets, using GEO accession #s GSE39152 (*Wakim et al., 2012*), GSE107281 (*Milner et al., 2017*), GSE70813 (*Mackay et al., 2016*), and GSE125471 (*Nath et al., 2019*). Normalized enrichment scores and false discovery rates were visualized using ggplot2 package in R.

## ChIP-Seq data analysis

Previously published ChIP-Seq data of in vitro activated CD8 T cells (Accession number GSE135533) were obtained from Gene Expression Omnibus (GEO) repository at the National Center for Biotechnology Information (NCBI). The data were mapped to the mouse reference genome (mm39) using Bowtie2. The SAM files were further processed, sorted, and filtered for uniquely mapping reads using Samtools. Peakcalling was performed using MACS2. Files in bedGraph format were converted to bigWig using bedGraphToBigWig program from University of California, Santa Cruz (UCSC) genome browser and visualized using integrative genomics viewer (IGV).

## RNA isolation for quantitative PCR analysis

Total RNA was extracted using Quick RNA MiniPrep Plus kit (Zymo Research). The quantity and quality of the RNA were analyzed using NanoDrop. Total RNA was reverse transcribed using M-MuLV reverse transcriptase (New England Biolabs) following the manufacturer's instruction. Quantitative real-time PCR was performed using 2× SYBR green master mix (Bimake). Relative gene expression was determined using ddCt method with Ribosomal protein L9 (rpl9) as reference. Sequences for primers are shown in the supplementary data (*Supplementary file 1*).

## Retroviral transduction

An MSCV-IRES-GFP (pMig)-EOMES vector (RV-EOMES-GFP) was used for ectopic gene expression in activated CD8 T cells (*Pearce et al., 2003*). Viruses were packaged in human embryonic kidney (HEK) 293 cells and viral supernatants were processed using Retro-X concentrators (Takara Bio. USA Inc). Naive CD8 T cells were purified by negative selection and stimulated in vitro with plate-bound anti-CD3/CD28. After 48 hr, activated CD8 T cells were spin infected (1800 × *g*, 60 min, 30°C) using concentrated supernatants and polybrene (1 μg/ml). After 3 hr, the washed cells were cultured in complete RPMI.

## Statistical analysis

To evaluate variability between biological replicates, p values were calculated using two-way ANOVA followed by Tukey's multiple comparison tests, using GraphPad Prism. Horizontal lines indicate comparisons between samples. For in vitro experiments, variability between technical replicates was evaluated using Student's *t* tests calculated using GraphPad Prism. Experiments were repeated two or three times, based on outcomes.

## Acknowledgements

Cells were purified with the assistance of the Flow cytometry Core Facility at UCONN Health. This work was funded by NIH grant AI123864 (LSC and SK) and a postdoctoral career development award from the American Association of Immunologists (KC).

## Additional information

### Funding

| Funder | Grant reference number | Author |
|---|---|---|
| National Institute of Allergy and Infectious Diseases | R01 AI123864 | Susan M Kaech |
| American association for Immunologists | AAI Careers in Immunology Fellowship | Linda S Cauley |
| University of Connecticut Health Center | bridge funding | Linda S Cauley |

The funders had no role in study design, data collection, and interpretation, or the decision to submit the work for publication.

### Author contributions

Karthik Chandiran, Conceptualization, Data curation, Formal analysis, Investigation, Methodology, Writing - original draft, Writing – review and editing; Jenny E Suarez-Ramirez, Yinghong Hu, Conceptualization, Data curation, Formal analysis, Investigation, Methodology, Writing – review and editing; Evan R Jellison, Formal analysis, Supervision, Writing – review and editing; Zeynep Ugur, Data curation, Formal analysis, Investigation, Methodology; Jun Siong Low, Bryan McDonald, Data curation, Formal analysis, Investigation, Methodology, Writing – review and editing; Susan M Kaech, Conceptualization, Resources, Funding acquisition, Project administration, Writing – review and editing; Linda S Cauley, Conceptualization, Funding acquisition, Writing - original draft, Project administration, Writing – review and editing

### Author ORCIDs

Karthik Chandiran (ID) http://orcid.org/0000-0003-2118-7946
Linda S Cauley (ID) http://orcid.org/0000-0001-9488-0341

### Ethics

Experiments were performed in accordance with protocol AP-200531-0824 approved by the UCONN Health Institutional Animal Care and Use Committee (IACUC). Every effort was made to minimize suffering.

### Decision letter and Author response

Decision letter https://doi.org/10.7554/eLife.76457.sa1
Author response https://doi.org/10.7554/eLife.76457.sa2

## Additional files

### Supplementary files
- Supplementary file 1. Primer sequences for qPCR.
- Supplementary file 2. p values for *Figure 1A, B*.
- Transparent reporting form

### Data availability

Sequencing data have been deposited in GEO under accession code GSE151637 Figure 3-source data 1 contain the numerical data used to generate the figures.

The following dataset was generated:

| Author(s) | Year | Dataset title | Dataset URL | Database and Identifier |
|---|---|---|---|---|
| Hu Y, Chandiran K, Cauley LS | 2021 | Transcriptional profile of Smad4-deficient CTLs is similar to TRM | https://www.ncbi.nlm.nih.gov/geo/query/acc.cgi?acc=GSE151637 | NCBI Gene Expression Omnibus, GSE151637 |

The following previously published datasets were used:

| Author(s) | Year | Dataset title | Dataset URL | Database and Identifier |
|---|---|---|---|---|
| Wakim LM, Woodward-Davis A, Liu R, Hu Y, Smyth GK, Bevan MJ | 2012 | Molecular signature of brain resident memory CD8+ T cells | https://www.ncbi.nlm.nih.gov/geo/query/acc.cgi?acc=GSE39152 | NCBI Gene Expression Omnibus, GSE39152 |
| Milner JJ, Goldrath AW | 2017 | RNA-Seq of CD8+ T cell subsets during LCMV infection | https://www.ncbi.nlm.nih.gov/geo/query/acc.cgi?acc=GSE107281 | NCBI Gene Expression Omnibus, GSE107281 |
| Shi W, Liao Y, Kallies A, van Gisbergen K | 2016 | Hobit and Blimp1 instruct a universal transcriptional program of tissue-residency in lymphocytes | https://www.ncbi.nlm.nih.gov/geo/query/acc.cgi?acc=GSE70813 | NCBI Gene Expression Omnibus, GSE70813 |
| Nath AP, Braun A, Ritchie SC, Carbone FR, Mackay L, Gebhardt T, Inouye M | 2019 | Comparative analysis reveals a role for TGF-β in shaping the residency-related transcriptional signature in tissue-resident memory CD8+ T cells | https://www.ncbi.nlm.nih.gov/geo/query/acc.cgi?acc=GSE125471 | NCBI Gene Expression Omnibus, GSE125471 |
| Wu B, Zhang G, Guo Z, Wang G, Xu X, Li J-L, Zheng J, Whitmire JK, Wan Yy | 2020 | SKI-dependent TGFb signaling dictates CD103 expression in CD8+ T cells (ChIP-seq) | https://www.ncbi.nlm.nih.gov/geo/query/acc.cgi?acc=GSE135533 | NCBI Gene Expression Omnibus, GSE135533 |

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
