## [Editor Report]

This manuscript provides important information concerning the impact of TGFB and SMAD factors on antiviral CD8 T cell differentiation and memory formation. The new and interesting data from in vivo performed experiments provide solid evidence in support of their claims. Overall, this is an important study with an impressive amount of data. The only weak point is the clarity of the overall goal/message of the study as stated in the introduction.

---

## [Decision Letter]

**Decision letter after peer review:**

[Editors’ note: the authors submitted for reconsideration following the decision after peer review. What follows is the decision letter after the first round of review.]

Thank you for submitting the paper "SMAD4 and TGFβ are architects of inverse genetic programs during fate-determination of antiviral memory CD8 T cells" for consideration by *eLife*. Your article has been reviewed by 3 peer reviewers, and the evaluation has been overseen by a Reviewing Editor and a Senior Editor. The reviewers have opted to remain anonymous.

Comments to the Authors:

We are sorry to say that, after consultation with the reviewers, we have decided that this work will not be considered further for publication by *eLife*.

Specifically, the authors are commended for their presentation of a rather extensive body of well performed experiments concerning reciprocal regulation of memory CD8 T cell development by SMAD4 and TGFbeta. Although most of the results support their conclusions, there is a critical absence of results from in vivo experiments that would yield results validating the in vitro observations. A potential impact of the different KO strains on naive T cell biology also needs to be investigated to obviate any possible bias towards differentiation of circulating Trm cells. While the gene expression regulation is of interest, the functional consequences of the regulation remain uncertain. For those reason, the current work has been considered as preliminary and it would require a serious effort (additional experiments) for it to be deemed meritorious for publication in *eLife*.

*Reviewer #1 (Recommendations for the authors):*

1. As the authors found SAMD4 regulated CD103 expression, and controlled the Trm/Tcm cells differentiation, was there any difference of Smad4 expression in the Tcm, CD69+CD103+ Trm, and CD69+CD103- Trm?

2. For most of times, the authors only showed percentage data, please include the cell number/magnitude of responses as well.

3. The author should clarify if the CD8 T cells in Figure 1 were tissue resident distinguished with i.v. antibody injection, for the mixed circulating and resident CD8 T cells may not be completely distinguished with surface marker expression levels.

4. The authors suggested that Smad4 altered homing receptor expression via a TGFb independent pathways by Figure 1A, however, I noticed that the DKO cells showed ameliorated features compared with the S4KO cells, and Figure 3 also suggested that TGFb signaling was involved in the phenotypes of S4KO cells.

5. Did the S4KO affect the effector function of CD8 T cells?

6. In the Figure 2F and 2G, the authors claimed that the EOMES expression downregulated in both spleen and lung, but T-bet rarely changed; however, I found the T-bet expression changed significantly in the lung, but EOMES didn't show much difference in the lung.

7. SMAD2/3 are the downstream of TGFbR2, however, in the Figure 3C, why S4TR2 DKO cells showed different features with S34 DKO?

8. Figure 3C and 3D showed the canonical TGFb pathways and SMAD4 cooperatively regulated CD103, was Eomes be regulated in the same way?

9. I suggest the authors to examine the effects of SMAD4 expression in CD8 T cells to provide the gain-of-function data.

*Reviewer #2 (Recommendations for the authors):*

The work is nicely performed. Enthusiasm is dampened by the high number of studies that are performed in vitro. For instance, T cells transduced with LV overexpressing transcription factors (e.g. Eomes) could be transferred and tracked in vivo.

There is a major concern on the impact of the different KO strains on naive T cell biology, i.e., whether the naive cells are impacted compared to control cells, for instance if they are already biased towards differentiation to circulating of Trm cells. More investigation in this regard (flow cytometry of several molecules, especially activation, or RNA seq) would be appreciated. Has TGFbRII KO an impact (e.g., autoimmunite traits) on the function of peripheral T cells?

Bar graphs should show values referred to single animals, not aggregate data, so to appreciate variability

Flow cytometry plots should have scales, so to check if gates are the same (for instance, it seems that different gates were used to define positive expression od CD103 in Figure 5B).

*Reviewer #3 (Recommendations for the authors):*

– The introduction summarizes a wide range of information, but does not appear to contain a specific research question or goal. On the second page of the intro, a sentence states "To further define how SMAD-dependent signaling pathway alter the migratory properties of pathogen-specific CTLs, we have used …". It seems like this might be the main aim of this study but migration/retention is never formally addressed. Further, the title seems to indicate the focus of the manuscript will be on differentiation of memory T cells, but much of the key experiments are performed in vitro with aCD3/CD28 activated cells. Specifying the goal of the research may help clarify the rationale and conclusions of the work.

---

## [Author Response]

[Editors’ note: The authors appealed the original decision. What follows is the authors’ response to the first round of review.]

Reviewer #1 (Recommendations for the authors):1. As the authors found SAMD4 regulated CD103 expression, and controlled the Trm/Tcm cells differentiation, was there any difference of Smad4 expression in the Tcm, CD69+CD103+ Trm, and CD69+CD103- Trm?

To address this concern, we used qPCR to compare SMAD4 expression in T_CM_ and T_RM_ cells (Figure 2 —figure supplement 1) and did not find a major difference in SMAD4 expression in these subsets. When other groups compared the transcriptional profiles of different subsets of pathogen-specific CTLs by RNA-sequencing, SMAD4 was not included on the list of differentially-expressed genes. Since SMAD4 chaperones phosphorylated RSMADs into the nucleus, dynamic changes in protein expression are not required to modify gene expression.

3. The author should clarify if the CD8 T cells in Figure 1 were tissue resident distinguished with i.v. antibody injection,

We did not use IV staining for the phenotypic analyses (Figures1A and 1B), however we used this technique to show that signaling via SMAD4 increases the numbers terminally-differentiated T_EFF_ cells that express KLRG1 and CX_3_CR1 in the vasculature (Figure 1C). The text and figure legends have been modified accordingly. P values for Figure 1 are shown in supplementary File 2.

4. The authors suggested that Smad4 altered homing receptor expression via a TGFb independent pathways by Figure 1A, however, I noticed that the DKO cells showed ameliorated features compared with the S4KO cells, and Figure 3 also suggested that TGFb signaling was involved in the phenotypes of S4KO cells.

This observation is correct. Our data show that TGFb collaborates with SMAD3 to induce CD103 expression in the absence of SMAD4 (Figure 2A). Consistently, SMAD4-deficient CTLs expressed CD103 at higher levels than S4TR2KO cells in vivo (Figure 1A) and CD103 expression on S4KO cells increased during in vitro stimulation with TGFb (Figure 3E). The text has been modified to further emphasize this point.

5. Did the S4KO affect the effector function of CD8 T cells?

We used the distal Lck promoter to prevent SMAD4 expression in peripheral CD8 T cells. In a prior study, we showed that SMAD4-deficient CTLs proliferated and expressed IFNγ and TNFa at similar levels as wildtype cells (Hu et al., 2015). In contrast, others reported an altered cytokine response after SMAD4 was ablated using the proximal Lck promoter (Cao *et al.,* 2015). This discrepancy may reflect a loss of signaling via SMAD4 during thymic development.

6. In the Figure 2F and 2G, the authors claimed that the EOMES expression downregulated in both spleen and lung, but T-bet rarely changed; however, I found the T-bet expression changed significantly in the lung, but EOMES didn't show much difference in the lung.

Our data show that EOMES is negatively-regulated by TGFb (Figure 4A in vitro and 4B in vivo). Since TGFb can be activated by viral neuraminidase, EOMES was expressed at low levels during IAV infection (Figure 4 —figure supplement 1). To minimize the influence of virally-induced TGFb, we performed similar experiments with LMOVA (Figures4B and 4C). The revised figure shows that CTLs express EOMES at reduced levels after SMAD4ablation (Figure 4C). Since we did not find a difference in T-bet expression by RNA-sequencing, the data have been removed.

7. SMAD2/3 are the downstream of TGFbR2, however, in the Figure 3C, why S4TR2 DKO cells showed different features with S34 DKO?

Our data show that S4TR2DKO and S34DKO cells express CD103 at higher levels than the controls (Figure 2A). Although both changes are statistically significant, the difference in CD103 expression is more pronounced in S4TR2DKO than S34DKO cells. We obtained similar results when CTLs were analyzed in vivo (Figure 2D). Several mechanisms could account for this difference including a feedback mechanism, or alternative TGFb receptor.

The difference was limited to CD103 expression, since S4TR2DKO and S34DKO cells expressed CD62L and KLRG1 at similar levels (Figure 2B and 2C). This point is discussed in the text.

8. Figure 3C and 3D showed the canonical TGFb pathways and SMAD4 cooperatively regulated CD103, was Eomes be regulated in the same way?

Yes, this question is correct. qPCR (Figure 4A) and flow cytometry (Figures4B and 4C) data both show that EOMES is down-regulated by TGFb and induced via SMAD4.

9. I suggest the authors to examine the effects of SMAD4 expression in CD8 T cells to provide the gain-of-function data.

As recommended, we used a retroviral vector to re-express SMAD4 in S4KO cells. CD103 expression decreased by approximately 50% when CTLs were analyzed after 6 days in culture (Figure 5 —figure supplement 5).

Reviewer #2 (Recommendations for the authors):The work is nicely performed. Enthusiasm is dampened by the high number of studies that are performed in vitro. For instance, T cells transduced with LV overexpressing transcription factors (e.g. Eomes) could be transferred and tracked in vivo.

For this study, a reductionist approach was used to reduce the numbers of variables that influence data interpretation. As requested, we provide new experiments showing CTLs analyzed in vivo (Figures 1B, 1C, 1D, 2C, 2D, 4F, 5D, Figure 1 —figure supplement 4, Figure 2 —figure supplement 1). The additional data include phenotypic analysis of OTI-S4KO cells analyzed after ectopic EOMES expression (Figure 5D). As expected, CD103 expression decreased after ectopic EOMES expression, which is consistent with our in vitro data (Figure 5E).

There is a major concern on the impact of the different KO strains on naive T cell biology, i.e., whether the naive cells are impacted compared to control cells, for instance if they are already biased towards differentiation to circulating of Trm cells. More investigation in this regard (flow cytometry of several molecules, especially activation, or RNA seq) would be appreciated.

We used the distal Lck promoter to induce gene-ablation at late stage of thymic development and found no evidence of defects during T cell priming. Our prior study shows that SMAD4-deficient CTLs proliferate and express IFNγ and TNFa at similar levels as wildtype cells (Hu et al., 2015). To further address this concern, we provide new data showing that OTI-STDKO and OTI-Ctrl cells proliferate at similar rates during IAV infection (Figure 1 —figure supplement 1). We also analyzed pathogen-specific CTLs during infection with LM-OVA (in vivo*)* using MHCI tetramers (Figure 2C and 2D). The data show that CTLs in S2KO and 3KO mice display a normal phenotype, whereas S23KO and ALK5KO cells resemble TR2Kos. These data support our conclusion that SMAD4 and TGFb alter homing-receptor expression via two independent signaling pathways.

Has TGFbRII KO an impact (e.g., autoimmunite traits) on the function of peripheral T cells?

This question was addressed previously, using two different promoters to prevent TGFbRII expression (Zhang *et al.,* Nat Immol 2012). When CD4-Cre was used for gene-ablation, spontaneous T cell activation caused fatal autoimmune disease within 3-5 weeks. Conversely, after the distal Lck-Cre promoter was used to prevent TGFbRII expression the mice did not develop autoimmune disease before the experiment was terminated at 18 weeks after birth. These data show that early gene-ablation impacts thymic development. Others recently found that autoimmune disease did not develop when CD4-Cre was used to produce CTLs with dual mutations (lacked both TGFbRII and SMAD4) Igalouzene *et al.,* (2022). Since SMAD4 and TGFb support normal thymic development, these studies underscore the importance using the distal Lck-Cre promoter to study peripheral CD8 T cells.

Bar graphs should show values referred to single animals, not aggregate data, so to appreciate variability

For this study we used P values and error bars representing SD to illustrate variability between samples. New data shown Figures2C and 2D are presented as scatter plots, with p values and SD as requested.

Flow cytometry plots should have scales, so to check if gates are the same (for instance, it seems that different gates were used to define positive expression od CD103 in Figure 5B).

The contour plots have been revised as requested.

Reviewer #3 (Recommendations for the authors):– The introduction summarizes a wide range of information, but does not appear to contain a specific research question or goal. On the second page of the intro, a sentence states "To further define how SMAD-dependent signaling pathway alter the migratory properties of pathogen-specific CTLs, we have used …". It seems like this might be the main aim of this study but migration/retention is never formally addressed. Further, the title seems to indicate the focus of the manuscript will be on differentiation of memory T cells, but much of the key experiments are performed in vitro with aCD3/CD28 activated cells. Specifying the goal of the research may help clarify the rationale and conclusions of the work.

The manuscript has been edited to improve clarity and emphasize the research goals*.*